# Post-Operative Endodontic Pain Management: An Overview of Systematic Reviews on Post-Operatively Administered Oral Medications and Integrated Evidence-Based Clinical Recommendations

**DOI:** 10.3390/healthcare10050760

**Published:** 2022-04-19

**Authors:** Federica Di Spirito, Giuseppe Scelza, Roberto Fornara, Francesco Giordano, Donato Rosa, Alessandra Amato

**Affiliations:** 1Department of Medicine, Surgery and Dentistry “Schola Medica Salernitana”, University of Salerno, Baronissi, 84081 Salerno, Italy; g.scelza4@studenti.unisa.it (G.S.); francescodrgiordano@gmail.com (F.G.); drosa@unisa.it (D.R.); aamato@unisa.it (A.A.); 2Private Practitioner, 20010 Milan, Italy; robertofornara01@gmail.com

**Keywords:** endodontics, non-steroidal anti-inflammatory drugs, pain, root canal treatment, steroidal anti-inflammatory drugs

## Abstract

Endodontic treatment comprises the overall management of pre-, intra- and post-operative symptoms, including post-operative endodontic pain, considered as a complication susceptible of chronicization. Post-operative pain is very common and highly unpreventable and has a multi-factorial etiology and a potential pathogenic link to the acute inflammation of the periapical area, secondary to localized chemical, mechanical, host and/or microbial damage occurring during endodontic treatment. Considering the multitude of heterogeneous technical and pharmacological approaches proposed to control post-operative endodontic pain, the present study primarily comprised an overview of systematic reviews of systematic reviews of randomized clinical trials, summarizing findings on post-operatively administered oral medications for post-operative endodontic pain control, in order to note the most effective type and dosage of such drugs. Secondarily, a narrative review of the current evidence on technical solutions to be observed during endodontic treatment procedures, to control post-operative pain, was conducted to provide integrated evidence-based clinical recommendations for optimal post-operative endodontic pain management.

## 1. Introduction

The majority of the patients visit the dental office because of more or less intense pain, mainly attributable to endodontic and periodontal causes, and most of all recognizing an endodontic origin; thus, it is essential for the clinicians to differentiate odontogenic from non-odontogenic pain [1]. Accordingly, endodontic pain management primarily depends on an accurate diagnosis of pain origin, aided by clinical exam and periapical and pulp tests, along with 2D and 3D radiographic examination [1,2].

Endodontic treatment comprises the overall management of pre-, intra- and post-operative symptoms, including post-operative endodontic pain, which is considered as a treatment complication susceptible of chronicization [3] and which represents patients’ significant concern [4].

Post-operative pain is very common, affecting from 2.5% to almost 60% of subjects that have undergone endodontic treatments [5], and it shows a tendency to increase between 6 and 12 h after treatment, reaching a prevalence of about 40% in 24 h and falling to 11% one week after treatment [3,6]. Moreover, post-operative endodontic pain is highly unpreventable, being affected by a variety of factors related to the subject (such as age and gender), to the treated tooth (including the pre-operative pulp status and type of tooth) and to the treatment performed (either primary root canal therapy or retreatment) [6].

Given that post-operative endodontic pain has a multi-factorial etiology and a potential pathogenic link to the acute inflammation of the periapical area, secondary to localized chemical, mechanical, host and/or microbial damage occurring during endodontic treatment [3,4,5,6], several strategies have been proposed to control post-operative endodontic pain, comprising technical features and solutions to be observed when performing endodontic treatment [5,7,8,9,10,11,12,13,14,15,16,17,18,19,20,21,22,23,24,25,26,27,28,29,30], along with occlusal reduction [31,32,33], as well as adjunctive therapies, such as lower-laser therapy [34,35], cryotherapy [36,37,38], phototherapy [39], topical medicaments administration and systemic drug therapy [6,40,41].

Considering the multitude of heterogeneous technical and pharmacological approaches proposed to control post-operative endodontic pain, the primary aim of the study was to summarize relevant systematic reviews of randomized controlled trials on post-operative oral medications administered in subjects that have undergone primary root canal therapy and retreatment, in order to note the type and dosage most effective in post-operative endodontic pain control. The secondary aim of the study was to integrate such findings with comprehensive evidence-based clinical recommendations for post-operative endodontic pain management, reviewing current evidence on technical solutions to be observed during endodontic treatment procedures for post-operative endodontic pain control.

## 2. Materials and Methods

The present study comprises an overview of systematic reviews of systematic reviews, summarizing findings on post-operatively administered oral medications for post-operative endodontic pain control in order to note the most effective type and dosage, and a narrative review of current evidence, highlighting technical solutions to be observed during endodontic treatment procedures to control post-operative pain in order to provide integrated evidence-based clinical recommendations for optimal post-operative endodontic pain management. Both the overview of systematic reviews and review protocols are described below.

### 2.1. Overview of the Systematic Reviews of Systematic Reviews Protocol

The overview of systematic reviews was conducted in accordance with the Preferred Reporting Items for Systematic Reviews and Meta-analyses (PRISMA) statement [42], which is freely available online (http://www.prisma-statement.org/, accessed on 28 September 2021), and the overview of systematic reviews protocol was developed before the analysis.

The focused question identified for the present overview of systematic reviews, as well as search strategies and selection criteria currently applied, were compliant with the PICO model [43] (https://linkeddata.cochrane.org/pico-ontology; accessed on 28 September 2021).

The question of the overview of systematic reviews was “Which type and dosage of oral medication administered after non-surgical initial treatment and retreatment is more effective to control post-operative endodontic pain?”, focusing on:

P—Population: subjects who completed non-surgical initial endodontic treatment or retreatment and experienced post-operative endodontic pain;

I—Intervention: post-operative administration of oral medications to treat post-operative endodontic pain;

C—Comparison: among different post-operatively administered types of oral medications and with placebo;

O—Outcome: reduction/resolution of post-operative endodontic pain.

#### 2.1.1. Overview of the Systematic Reviews Search Strategy

The Cochrane Database of Systematic Reviews and MEDLINE/PubMed were electronically searched for systematic reviews with or without meta-analyses until 30 September 2021, by two independent reviewers (F.D.S. and G.S.), using the following search terms combined by Boolean operators:endodontic pain OR post-endodontic pain OR postendodontic pain OR endodontic post-operative pain OR endodontic postoperative pain.
AND
2.oral medicines OR oral drugs OR oral medications OR oral medicaments OR oral administration OR orally administered medicines OR orally administered drugs OR orally administered medications OR orally administered medicaments OR treatment OR therapy OR management OR approach OR strategies.
AND
3.Systematic Review OR Meta-analysis.

#### 2.1.2. Overview of the Systematic Reviews Study Selection and Eligibility Criteria

Retrieved citations were recorded, and duplicates were eliminated. Titles and abstracts of all the studies initially identified through the electronic search were independently screened by the two reviewers (F.D.S. and G.S.). Full-texts were also independently reviewed for ambiguous abstracts and for articles considered appropriate to the purpose of the present study and potentially eligible for the present overview of systematic reviews, according to the inclusion and non-inclusion criteria, reported in Table 1.

Disagreements between reviewers were solved through discussion and consensus.

Reference lists of included articles were also manually searched for additional potentially relevant systematic reviews with or without meta-analyses; identified titles and abstracts were screened, and full-texts were reviewed, as described above. In case of missing information or full-text unavailability, no attempt to contact authors was performed.

#### 2.1.3. Data Extraction of the Articles Included in the Overview of Systematic Reviews

Data extraction was independently conducted through a standardized data extraction form by the two reviewers (F.D.S. and G.S.), who solved disagreements by discussion and consensus.

The following data were recorded for each of the systematic reviews with or without a meta-analysis included in the present overview of systematic reviews: authors, year and journal of publication; trial’s number; characteristics of the trial’s population concerning the total number, age and gender of enrolled subjects; characteristics of the non-surgical endodontic treatment received (initial or re-treatment); post-operative endodontic pain assessment method and related follow-up period; characteristics of the trial’s intervention and comparison regarding the type, dosage and duration of post-operatively administered oral medications and a comparison against drugs or a placebo; trials’ statistically significant results concerning post-operative endodontic pain control.

#### 2.1.4. Quality Assessment and Data Synthesis of the Articles Included in the Overview of Systematic Reviews

Included systematic reviews were assessed for quality through the Assessing the Methodological Quality of Systematic Reviews 2 (AMSTAR 2) too [43], freely available online (https://amstar.ca), accessed on 29 September 2021.

Two authors (F.D.S., G.S.) qualitatively synthesized the characteristics of the trial’s population, intervention and comparison and reported significant results of each systematic review included in the present overview of systematic reviews.

### 2.2. Review Protocol

The narrative review question, also compliant with the PICO model [44], was “Which technical features and solutions to be observed during non-surgical initial endodontic treatment or retreatment procedures are more effective to control post-operative endodontic pain?”, identifying:

P—Population: subjects who completed non-surgical initial endodontic treatment or re-treatment;

I—Intervention: technical features and solutions to be observed during primary root canal treatment or retreatment procedures;

C—Comparison: among different technical features and solutions;

O—Outcome: reduced post-operative endodontic pain incidence, severity and duration.

#### Review Search Strategy and Eligibility Criteria

The narrative review was accomplished through a further electronic literature search, independently performed by two reviewers (F.D.S. and G.S.).

MEDLINE/PubMed and the Cochrane Database of Systematic Reviews were searched for trials and observational studies, as well as for systematic reviews, with or without meta-analyses, reviews and book chapters, until 30 September 2021, employing the following same search terms:endodontic pain OR post-operative endodontic pain OR postendodontic pain OR endodontic post-operative pain OR endodontic postoperative pain.
AND
2.endodontic treatment OR procedure OR therapy OR irrigation OR instrumentation OR obturation.

Studies investigating pre- and post-operative, either local or systemic, pharmacological approaches were excluded from the review, as well as those reporting adjunctive therapies, such as topical medicaments administration, lower-laser therapy, cryotherapy, phototherapy and similar.

## 3. Results

### 3.1. Overview of Systematic Reviews: Study Selection

A total of 99 potentially relevant titles/abstracts were obtained from the literature search, of whom 93 were from MEDLINE/PubMed, and six were from the Cochrane Database of Systematic Reviews. One duplicate was removed, and after the titles/abstracts screening, 58 records were excluded, specifically 57 because they were not pertinent and 1 because no full-text was available; consequently, 40 potentially eligible full-text articles were assessed. Based on the eligibility criteria reported in Table 1, seven systematic reviews also including observational or preclinical or in vitro studies besides RCTs, 21 concerning technical features and other therapies reducing post-operative endodontic pain, four studies evaluating pre-operatively administered oral medications and one also comprising antibiotic prescription were not considered in the present analysis; one study evaluating non-odontogenic pain was removed, and one article was excluded because no outcomes data could be extracted.

Finally, five studies were included in the present overview of systematic reviews [4,45,46,47,48]. Figure 1 shows the study selection flow-chart.

### 3.2. Overview of Systematic Reviews: Studies’ Characteristics and Synthesis of the Reported Results

Out of the five systematic reviews included in the present overview of systematic reviews of systematic reviews, all considered exclusively RCTs and were published in the English language, four [4,46,47,48] also comprised meta-analyses and only one [47] declared funding.

Findings from a total of 44 trials, accounting for 2697 enrolled subjects between 15 and 68 years of age, were retrieved, all concerning completed non-surgical initial endodontic treatments, performed in single [47], single/multiple [4,46] or an unspecified [45,48] number of sessions, of both vital and non-vital teeth and with none reporting retreatments.

Post-operative endodontic pain was assessed at different time points through Visual Analogue Scales in four studies [4,45,47,48], while in one study, the method was not specified [46]; the follow-up period ranged from 1 h to 7 days after the endodontic treatment.

The type and dosage of the oral medications, post-operatively administered to control post-operative endodontic pain, as well as the comparison with drugs and/or a placebo were clearly reported in all the studies, while duration was never specified. Four studies [45,46,47,48] evaluated the efficacy of non-steroidal anti-inflammatory drugs (NSAIDs).

A detailed summary of the data recorded for each study is shown in Table 2, reporting the authors, year and journal of publication; the trial’s number; characteristics of the trial’s population (subjects number, age and gender) and of the endodontic treatment received (non-surgical initial endodontic treatment or retreatment of vital or non-vital teeth and the number of sessions); post-operative endodontic pain assessment method and follow-up period; characteristics of trial’s intervention and comparison (type, dosage and duration); the trial’s significant results.

No meta-analysis could be performed due to the high heterogeneity and incompleteness of the retrieved data.

### 3.3. Narrative Review: Study Screening Process and Included Studies

From literature research, a total of 127 articles were retrieved from MEDLINE/PubMed and six from the Cochrane Database of Systematic Reviews, and no duplicate elimination was necessary. After titles/abstract screening, 52 articles were removed because of not being pertinent. Full-text reading of potentially relevant articles was performed for 75 records, specifically 40 trials and observational studies, 24 systematic reviews (of whom 9 were with meta-analyses) and 11 narrative reviews. Finally, 22 systematic reviews with or without meta-analyses relevant to the topic and to the aim of providing clinical recommendations to control post-operative endodontic pain were obtained [5,22,23,24,25,26,27,28,29,30,31,32,33,34,35,36,37,38,39,40,41,49] along with 26 among trials, observational studies and reviews [7,8,9,10,11,12,14,15,16,17,18,19,20,21,50,51,52,53,54,55,56,57,58,59,60,61,62] that were compliant with the eligibility criteria.

## 4. Discussion

Several technical features and solutions [5,7,8,9,10,11,12,13,14,15,16,17,18,19,20,21,22,23,24,25,26,27,28,29,30,31,32,33,34,35,36,37,38] as well as systemic post-operative pharmacological therapies [4,6,40,41,46,47,48] have been proposed to control post-operative endodontic pain. Given the multi-factorial etiology of such a complication, a comprehensive approach, integrating both aspects, may be more effective and, therefore, recommended in the clinical management of post-operative endodontic pain. Accordingly, the current study presented an overview of systematic reviews of systematic reviews on oral medications post-operatively administered in subjects who completed either non-surgical endodontic initial treatment or retreatment, comparing them orally with each other and with a placebo, to note the most effective type and dosage for post-operative endodontic pain control and a narrative review, highlighting recent evidence on technical solutions to be observed during endodontic treatment procedures to control post-operative pain, in order to provide clinicians with integrated clinical recommendations for optimal post-operative endodontic pain management.

### 4.1. Oral Medications Post-Operatively Administered to Control Post-Operative Endodontic Pain

Previous systematic reviews have mostly analyzed single classes of oral medicaments, including steroidal (SAIDs) and non-steroidal anti-inflammatory drugs (NSAIDs), as well as opioid analgesics, not comparing one another and, thus, failing to provide conclusive results on the superiority of one over another approach. In addition, specific data regarding indications, timing and dosage of the post-operative endodontic oral medications prescription were often not clearly presented, thus not providing clinically relevant recommendations. Moreover, several reviews also evaluated antibiotics and pre-medications to control post-operative endodontic pain, which were both excluded from the present overview of systematic reviews of systematic reviews, because they contrasted with antibiotic stewardship standards proposed both by the American Dental Association [63] and the American Association of Endodontists [64] and with the attempt of avoiding polypharmacy [64], respectively, and they should, therefore, be considered not clinically advisable.

Systematic reviews reporting findings from subjects with cancer, osteonecrosis of the jaws or other systemic disorders possibly affecting pain perception or the efficacy of the investigated oral medications [65,66,67,68] were also not considered in the attempt to obtain generalizable data.

No data were retrieved differentiating vital from non-vital teeth, non-surgical initial endodontic treatment from retreatment and sessions number, neither in terms of post-operative endodontic pain incidence and severity nor management; thus, no specific recommendations could be formulated concerning pre-operative pulp conditions, the type of non-surgical endodontic treatment and single- vs. multiple-sessions procedures.

However, with regard to the type and dosage of post-operatively administered medications, steroidal anti-inflammatory drugs were shown to be effective in reducing pain intensity in the first 24 h after a primary root canal treatment. SAIDs’ analgesic efficacy was shown to be potentially influenced by the type and dose of the drug; no significant difference was nonetheless reported between dexamethasone (4 mg) and betamethasone (2 mg) [31].

Nonsteroidal anti-inflammatory drugs were the most common oral medicaments post-operatively administered to control pain, with ibuprofen being the most prescribed and investigated. Ibuprofen (600 mg) alone and ibuprofen (600 mg) combined with acetaminophen (1000 mg) were reported to be significantly more effective in post-operative endodontic pain control when administered 6 h after endodontic treatment and may be, therefore, recommended as a first-choice treatment in the first hours following non-surgical endodontic treatment. In addition, emerging data support the evidence that ketoprofen (50 mg) as well as naproxen (500 mg) might be even more effective than ibuprofen (600 mg) alone 6 h post-operatively [47]. However, only low to moderate evidence supports such a recommendation [48]. Moreover, multiple-dose regimens seemed to be more effective in pain control compared to single-dose, although no definite protocols were proposed [45].

The main limitation of this overview of systematic reviews of systematic reviews is the small number of included studies, ascribable to the restricted eligibility criteria, fitting the PICO-focused question. Other limitations include variability in post-operative endodontic pain assessment time points, excluding the possibility to quantitatively synthesize and compare the results from the included studies; a lack of information regarding indications and, especially, duration of the oral pharmacological therapies, perpetuating the empirical approach to oral drugs prescription to control post-operative endodontic pain and significant heterogeneity of the included systematic reviews, principally concerning the type and dosage of the post-operatively administered oral medications, which may not all be available at the same concentrations and combinations worldwide, thus reducing conclusions’ generalizability.

However, to the authors’ knowledge, this study was the first summarizing findings from systematic reviews of RCTs on different classes of oral drugs prescribed to control post-operative endodontic pain all together. In addition, the presented overview of systematic reviews of systematic reviews did not consider either pre-operatively administered oral medications or combined antibiotic prescriptions in the attempt to highlight the shorter therapy duration, thus excluding pre-medications, and to avoid unnecessary antibiotic treatment to control post-operative endodontic pain. Moreover, we aimed to identify the most effective type and dosage of post-operatively administered medications, not only limiting, at the most, therapy duration and complying with accepted standards for antibiotic stewardship, but also taking into account polypharmacy, which is especially relevant for patients’ comorbidity and specific conditions [48,69,70,71,72,73,74]. In such a perspective, the correct pharmacological strategy should also reduce patients’ concerns, on the one side, and optimize practitioners’ time and efforts, on the other side, thus decreasing endodontic complications and emergencies, which may potentially require additional operative sessions, which should be even more avoided in the ongoing context of COVID-19 [75,76,77,78,79].

### 4.2. Evidence-Based Clinical Recommendations for Post-Operative Endodontic Pain Management

Pain following endodontic treatment may be secondary to anatomical factors, such as apical foramen position, pulp tissue localization in areas where it cannot be removed without proper care and root canals that can be omitted without instrumentation, as well as to technical factors affecting or occurring during the procedure, such as an inaccurate determination of the working length. The last could, in turn, lead to excessive instrumentation, with extrusion of root canal debris beyond the apical foramen during instrumentation, an extension of the filling beyond the apex and an incorrect use of irrigants, specifically sodium hypochlorite and hydrogen peroxide, potentially causing iatrogenic periapical discomfort [50,51]. Mechanical and chemical damages to the periapical tissues would also determine an inflammatory reaction that causes pain. The severity of the pain has been reported to be dependent on several aspects, including the intensity of the injury, the intensity of the tissue damage and the inflammatory response’s intensity. The mechanical irritation that causes periapical inflammation includes over-instrumentation, irritants and drugs over the apical foramen and excessively extensive filling materials. Furthermore, the greater the amount of overextended material, the greater is the intensity of damage to the periapical tissues [62].

Pain after root canal treatment is a serious health problem that affects the quality of life in the short term and, sometimes, even in the long term. Root canal treatment is generally very effective in relieving tooth pain [50,52,53] when correctly performed. However, it has been reported that one in five patients, on average, receiving root canal treatment will experience a serious disturbance in their daily life due to post-operative endodontic pain [50] and that up to 10% of patients may refer to persistent pain until six months after endodontic treatment [51]. Indeed, endodontic therapy does not rely on biological consequences solely, but also on minimizing patient discomfort. The success of endodontic treatments is based on eliminating microbes from the root canal and on creating a local environment favoring healing, by performing appropriate shaping, 3D cleaning and, finally, a 3D obturation [5,7,8,9,10,11,12,13,14,15,16,17,18,19,20,21,22,23,24,25,26,27,28,29,30,54,55].

Silva et al. compared rotary instruments with hand files and concluded that a lower incidence of post-operative pain was recorded when rotary systems were employed [56]. Furthermore, these authors highlighted that rotary systems may reduce the overall risk of complications throughout the procedure, favoring both less debris formation and bacteria extrusion [56].

Based on the evidence, no significant differences in post-operative endodontic pain incidence, severity or control capability were reported for a single session when compared to multiple sessions endodontics [57,58]. However, numerous studies have also reported inter-appointment pain, putatively secondary to the imbalance in the host–bacteria relationship, induced by intra-canal procedures [5]. In such particular circumstances, pain may be due to the pathogenic effect of specific bacteria, such as *Porphyromonas Endodontalis*, *Porphyromonas Gingivalis* and *Prevotella species*, as proposed by various studies [59,60,61], or it may be linked to an individual’s host resistance [60]; it has been hypothesized that subjects who have a lesser ability to cope with infections may be more prone to develop clinical symptoms after endodontic procedures in infected root canals. Calcium hydroxide is considered an optimum root canal medication against microbes in multiple sessions treatments; such intra-canal medicaments have been found to be able to reduce the periapical infection, both decreasing the total count of those microbes present in the root canals and retarding the growth of new pathogens [49]. In addition to intracanal medications, several methods have been proposed to control pain occurring during endodontic treatment and resulting from inter-appointment exacerbations, comprising re-instrumentation, chirurgical incision, abscess drainage, occlusal reduction and systemically delivered medications [62].

Given that post-operative pain is very common and highly unpreventable [47], patients should always be advised that pain may be felt after the procedure, and considering the multi-factorial etio-pathogenesis of post-operative endodontic pain, practitioners should always steer clear of iatrogenic injury. Indeed, dentists should respect the working length during shaping, cleaning and obturation, both during root canal treatment and retreatment, accomplishing a correct access cavity to find all the root canals, using a crown down technique and respecting the apical third, avoiding debris, irrigants, gutta-percha and sealer extrusion over the apex and, finally, performing a correct obturation.

## 5. Conclusions

Post-operative endodontic pain remains still often undervalued, although it may lead, if not correctly managed, to both local and central sensitization, with pain chronicization [80,81,82,83].

For effective endodontic pain management, the causative factors and treatment methods discussed above should be given due consideration to help the patient get rid of the unpleasant feeling of pain.

Therefore, patients should always be advised that pain may be felt after the procedure, and practitioners should, beyond correctly identifying odontogenic pain, diagnose its causes and, if possible, adapt clinical procedures according to the evidence-based findings. Moreover, the described clinical protocols, optimizing post-operative endodontic pain control, should be integrated with the presented recommendations on post-operatively administered oral medicaments, proposing the use of ibuprofen (600 mg) alone and combined with acetaminophen (1000 mg) as a first choice treatment, or ketoprofen (50 mg) as well as naproxen (500 mg) 6 h post-operatively [47], also in the perspective of minimizing drug intake, as it may be especially relevant in fragile and elderly subjects.

## Figures and Tables

**Figure 1 healthcare-10-00760-f001:**
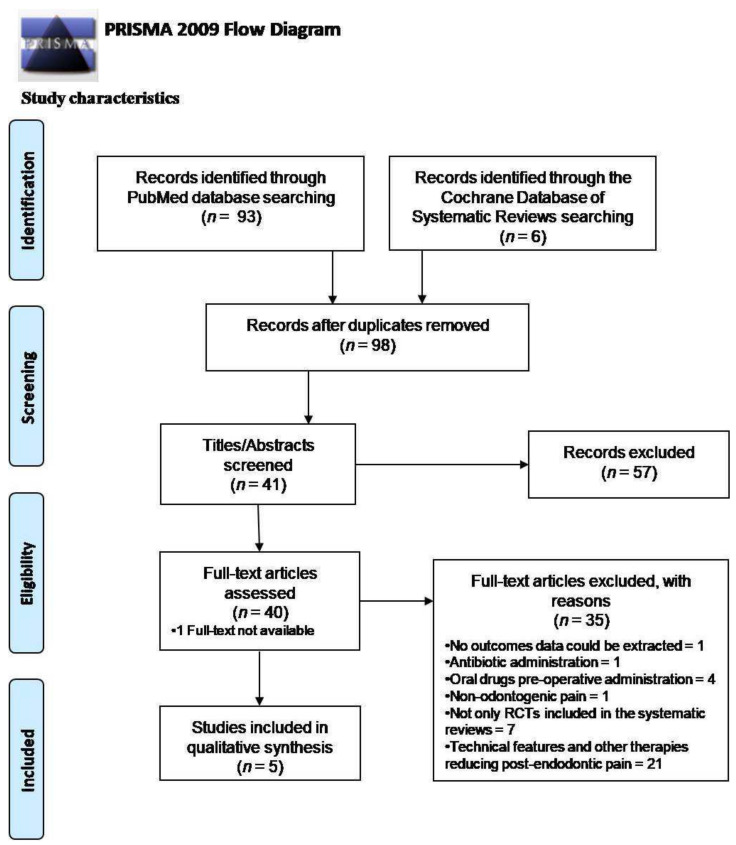
This is a figure. PRISMA flow chart of the study.

**Table 1 healthcare-10-00760-t001:** This is a table. Systematic reviews’ eligibility criteria concerning sources, design and characteristics of the studies included regarding population, intervention, comparison and outcome(s).

Systematic Reviews Eligibility	Inclusion Criteria	Non-Inclusion Criteria
Sources Databases Language Publication status Publication date Text availability	Electronic and Manual English language Published or in press or ahead of print No restrictions Full-text access	None Non-English language Submitted No restrictions Abstractonly
Design	Systematic Reviews exclusively including Randomized Clinical Trials (RCTs) with or without a meta-analysis	Systematic Reviews also including ;prospective, retrospective, case-control, cross-sectional, case series, case reports or pre-clinical in vivo in vitro studies, as well as conference papers, oral communications, books and chapters
Characteristics of the studies included in the eligible systematic reviews RCTs Population Study sample size Age Gender Treatments	Subjects who completed endodontic treatment or retreatment with post-operative endodontic pain No restrictions Adulthood Old age No restrictions Non-surgical endodontic initial treatment and retreatment	Subjects who did not complete endodontic treatment or retreatment Subjects with periodontal pain Subjects with endodontic–periodontal lesions Subjects with disorders altering either pain perception or post-operatively administered oral medications effectiveness in post-operative endodontic pain control Subjects with oral cancer and osteonecrosis of the jaw Subjects with dental phobia No restrictions Childhood No restrictions Surgical endodontic treatment
Intervention Route of administration Timing of administration Type of medicaments	Pharmacological therapy Oral Post-operative administration Anti-inflammatory drugs (any) Analgesics (any)	Other therapies or treatments Parenteral Topical (intracanal, intraligamentary, supraperiosteal, intraosseous) Pre-operative administration In-between sessions administration Antibiotics
Comparison	Placebo use Other type of medicaments	Combined premedication and post-operative pharmacological pain treatment
Outcome(S) Pain evaluation method	Post-operative endodontic pain control Visual Analogue Scale (VAS)	Pre- and intra-operative endodontic pain control Periodontal pain control

VAS: Visual Analogue Scale; RCT: Randomized Clinical Trials.

**Table 2 healthcare-10-00760-t002:** Characteristics of the included systematic reviews and summary of the reported statistically significant results on the efficacy of post-operative endodontic pain control of post-operatively administered oral medications.

Intervention Oral Medication Type Dosage Duration Comparison Placebo Others	Author, Year Reference [] Journal Funding	Trials *n*	Population N. F/M Age (y.o.)	Characteristics of the Non-Surgical Endodontic Treatment Received Vital and Non-Vital Teeth Primary Root Canal Therapy/Retreatment N. of Sessions	Method to Assess/Grade post-Operative Pain VAS Others Follow-Up Period	Outcome(s) Statistically Significant (*p* < 0.05) Pain Reduction
SAIDs Betamethasone 2 mg Dexamethasone 4 mg Dexamethasone 0.75 mg Duration: n.a. Comparison: placebo	Shamszadeh, 2018 [31] J Endod No Funding	4	166 F/M: n.a. >15 y.o.	Vital and Non-vital teeth Primary root canal therapy N. of sessions: single and multiple	VAS Follow-up period for Dexamethasone: 6, 24 h Follow-up period for Betamethasone: 6, 12, 24 h	Pain scores at 6 h: SAIDs <placebo (95% CI, −1.55 to −0.51; *p* = 0.000) Pain scores at 12 h: SAIDs <placebo (95% CI, −1.71 to −0.46; *p* = 0.001) Pain scores at 24 h: SAIDs <placebo (CI, −1.34 to −0.56; *p* = 0.000)
NSAIDs Ibuprofen 400 mg Ibuprofen 400 mg + alprazolam 0.5 mg Ketorolac 10 mg Flurbiprofen 50 mg Naproxen 500 mg Ibuprofen 200 mg + paracetamol 325 mg + 40 mg caffeine Meloxicam 15 mg Piroxicam 20 mg Ibuprofen 600 mg Salicylic acid 650 mg Paracetamol 650 mg Ketoprofen 50 mg Paracetamol 325 mg + codeine 60 mg Ibuprofen 600 mg + paracetamol 1000 mg Duration: n.a Comparison. placebo Ibuprofen 400 mg Ibuprofen 400 + Alprazolam 0.5 mg Flurbiprofen (Loading dose 100 mg, subsequent 50 mg) Tramadol (Loading dose 100 mg subsequent 100 mg) Flurbiprofen + tramadol (Loading dose 100 mg subsequent 50/100 mg) Paracetamol 1000 mg Ibuprofen 600 + paracetamol 1000 mg Mefenamic acid 500 + paracetamol 1000 mg Diclofenac K 50 + paracetamol 1000 mg Dexamethasone 4 mg Dexamethason 0.75 mg Ibuprofen + paracetamol 400/325 mg DiclofenacSodium + paracetamol 50/500 mg Tramadol 100 mg Paracetamol 325+ ibuprofen 200+ caffeine 40 mg Naproxen 500 mg Ibuprofen 600 mg Pentazocine + naloxone 50/0.5 mg Indomethacin 75 mg Ibuprofen 150 mg Ibuprofen 150 + paracetamol 250 mg Duration: n.a. Comparison: placebo	Smith, 2017 [47] J Endod Funding: Department of Endodontology Les Morgan Endowment and resident research grant from the American Association of Endodontists Foundation Zanjir, 2020 [48] J Endod No funding	9 11	499 F/M: n.a. 18–80 y.o. 706 F/M. n.a. 18–68 y.o.	Vital and Non-vital teeth Primary root canal therapy N. of sessions: single Vital and Non-vital teeth Primary root canal therapy N. of sessions: n.a.	VAS Follow-up period for Alprazolam, Naproxen, Ibuprofen, Salicidic acid: baseline, 6, 12, 24 h Follow-up period for Ketorolac, Flurbiprofen: baseline, 6, 24 h Follow-up period for Piroxicam, Ibuprofen: baseline, 24 h 10 point, 100 mm, 170 mm Heft Parker VAS Numerical rating scale 0 to 11 Follow-up period for Ibuprofen 400 mg, Ibuprofen 400 + Alprazolam 0.5 mg: 6, 12, 24, 48, 72 h Follow-up period for Flurbiprofen:6, 24, 48 h Follow-up period for Paracetamol 1000 mg Ibuprofen 600 + paracetamol 1000 mg Mefenamic acid 500 + paracetamol 1000 mg: 1, 2, 3, 4, 6, 8 h Follow-up period for DiclofenacSodium + paracetamol 50/500 mg Tramadol 100 mg Paracetamol 325+ ibuprofen 200+ caffeine 40 mg Naproxen 500 mg: 6, 12, 24 h [14,45] Follow-up period for Dexamethasone 4 mg and 0.75 mg: 8, 24, 48 h Follow-up period for Naproxen 500 mg Ibuprofen 600 mg: 0, 6, 12, 24, 48 h Follow-up period for Ibuprofen 150 mg Ibuprofen 150 + paracetamol 250 mg: 0 to 5 days	Pain scores at 6 h: Ibuprofen 600 mg < placebo (ES = 10.50, *p* = 0.037) Pain scores: Ibuprofen 600 mg + paracetamol 1000 mg < placebo (ES = 34.89, *p* = 0.000) Pain scores at 6–8 h: NSAIDs + paracetamol< placebo(MD = −22; 95% CrI = −38 to −7.2) NSAIDs < placebo (MD = −21; 95% CrI = −34 to −7.6) Pain scores at 12 h: Only NSAIDs were effective in decreasing postoperative pain (MD = −28; 95%CrI = −49 to −7) Pain scores at 24 h: Only NSAIDs were effective in decreasing postoperative pain (MD = −15; 95%CrI = −27 to −2.3) Pain scores at 48 h: No pain reduction
SAIDs NSAIDs and Paracetamol Opioid analgesics Tramadol 100 mg Flurbiprofen 50 mg every 6 h for 2 days Flurbiprofen 50 mg + tramadol 100 mg Ibuprofen 600 mg single dose Ibuprofen600 mg + paracetamol 1000 mg Paracetamol 325 mg + ibuprofen 200 mg + caffeine 40 mg every 6 h for 2 days Naproxen 500 mg every 6 h for 2 days Ibuprofen 400 mg every 6 h for 2 days Alprazolam 0.5 mg + ibuprofen 400 mg Tramadol37.5 mg +paracetamol 325 mg every 4 h for 3 days Duration: n.a. where not specified Comparison: placebo, codeine + paracetamol, each other	Santini, 2020 [45] Endod J No funding	5	266 18–68 y.o.	Vital and Non vital teeth Primary root canal therapy N. of sessions: n.a.	VAS Follow-up period for Ibuprofen (600 mg): 0, 1, 2, 3, 4, 6, 8 h Follow-up period for paracetamol + ibuprofen + caffein, naproxen: 6, 12, 24 h Follow-up period for Ibuprofen (400 mg), alprazolam + ibuprofen Follow-up period for tramadol + paracetamol: 0, 6, 12, 24, 48, 72 h	Pain decreased in all groups over time Pain scores at 1 h: Reduction in pain scores in all groups (*p* < 0.001) At 6, 12 and 24 h, pain was lower in the experimental groups than in the placebo (*p* < 0.01) Pain scores at 4 h: Alprazolam + ibuprofen < other groups (*p* < 0.0001) Ibuprofen/paracetamol < placebo (*p* < 0.001) Pain scores at 6 h: Ibuprofen + alprazolam< Ibuprofen (*p* = 0.018) and placebo (*p* = 0.018) Pain scores at 8 h: Ibuprofen/paracetamol < placebo (*p* < 0.001) Pain scores at 12 h: Ibuprofen + alprazolam < placebo (*p* < 0.001)
NSAIDs and paracetamol Ibuprofen Meloxicam Piroxicam Diclofenac sodium Tramadol Novafen Naproxen Indomethacin Diclofenac sodium/Paracetamol Paracetamol/Ibuprofen Ibuprofen/Alprazolam Rofecoxib Etodolac Celecoxib Dosage: n.a. Duration: n.a. Comparison: placebo	Shirvani, 2017 [46] J Oral Rehabil No funding	15	1060 >15 y.o.	Vital and Non-vital teeth Primary root canal therapy N. of sessions: single and multiple (2)	Method to assess/grade post-operative pain: n.a. Follow-up period: 0, 6, 12, 24 h	Pain scores immediately after the procedure: non-narcotic analgesics < placebo (MD of −0.50; 95% CI = −0.70, −0.30; *p* = 0.000) Pain scores at 6 h: non-narcotic analgesics < placebo (MD of −0.76; 95%CI = −0.95, −0.56; *p* = 0.000) Pain scores at 12 h: non-narcotic analgesics < placebo (MD of −1.15; 95% CI = −1.52, −0.78; *p* = 0.000) Pain scores at 24 h: non-narcotic analgesics < placebo (MD of −0.65; 95% CI = −1.05, −0.26; *p* = 0.001)

SAIDs = Steroidal anti-inflammatory drugs; NSAIDs = Non-steroidal anti-inflammatory drugs; n.a. = not available; VAS = Visual Analogue Scale; h = hours; MD = mean difference; EF = Effect size; Crl 95% = credible intervals, including with 95% probability that the treatment effect was evaluated in pairs in new trials.

## Data Availability

Medline (PubMed) and Scopus databases.

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
