# Peer review of "Post-Operative Endodontic Pain Management: An Overview of Systematic Reviews on Post-Operatively Administered Oral Medications and Integrated Evidence-Based Clinical Recommendations"

_healthcare, 2022, doi:10.3390/healthcare10050760_

Round 1

Reviewer 1 Report

What do you understand by:

"pertinent systematic reviews" 

"articles were removed because not pertinent." 

Did you take into consideration also pain from endodontically treated teeth under orthodontic treatment? 

Due to the limitation of the study, generated by the small number of included articles, have you considered to enlarge the data? Or the time period of the published included articles?

Why use antibiotics for pain control: "antibiotic treatment to control post-operative endodontic pain" ? 

This is a suposition: "potentially requiring additional operative ses-296 sions, that should be even more avoided in the ongoing context of the COVID-19" Covid 19 has nothing to do with pain evolution. 

The practitioner intends to perform the root vanal treatment correctly.... :"when correctly performed"  So, if pain emerges, it is not iatrogenically  intended. 

Conclusions should be more precise. They are ambiguous. 

Re-consider "For effective endodontic pain management, the causative factors and treatment 358 methods discussed above should be given due consideration to help the patient get rid of the unpleasant feeling of pain." "Presented recommendations" -  which? 

"fragile and elderly subjects" were not the subject of the research - so how can you conclude this? 

Author Response

On behalf of the Authors, I would first like to express our appreciation for the time and efforts spent to review our manuscript and for the comments and suggestions offered by the Referee and apologize for the difficulties encountered.

We welcome the suggestions for improving manuscript readability and we have tried to address the concerns expressed.

The authors have made changes to the manuscript according to the valuable comments and suggestions and to assist you in your work, we have highlighted all changes to the manuscript in yellow and we have reported the point-to-point replies to questions, concerns and suggestions below.

Table 1 and Table 2 have been edited and correctly located in the text.

We sincerely hope that the man script is now suitable for publication.

Kind regards

Dr. Federica Di Spirito, DDS, PhD

Concern of the Referee: What do you understand by: "pertinent systematic reviews" "articles were removed because not pertinent." 

Our response: With the term “pertinent” we meant appropriate to the topic and to the purpose of the study; we have substituted such term with “relevant” and “appropriate to the purpose of the present study” throughout the manuscript.

Concern of the Referee: Did you take into consideration also pain from endodontically treated teeth under orthodontic treatment? 

Our response: Inclusion criteria concerning population investigated in the studies included in the systematic reviews, considered in the present overview, described in Table 1, were: Adult subjects who completed endodontic treatment or retreatment with post-endodontic pain, with no age nor sample size restrictions. Exclusion criteria, instead, were: Subjects who did not completed endodontic treatment or retreatment, Subjects with periodontal pain, Subjects with endodontic-periodontal lesions, Subjects with disorders altering either pain perception or post-operatively administered oral medications effectiveness in post-endodontic pain control, Subjects with oral cancer and osteonecrosis of the jaws, Subjects with dental phobia. Therefore, orthodontic subjects were also potentially eligible, although no study specifically evaluated pain from endodontically treated teeth under orthodontic treatment.

Concern of the Referee: Due to the limitation of the study, generated by the small number of included articles, have you considered to enlarge the data? Or the time period of the published included articles?

Our response: We especially thank you for the consideration and, in accord with your observation regarding the small number of included articles, we have stated in the manuscript, along with other study limitations, “The main limitation of this overview is the small number of included studies, ascribable to the restricted eligibility criteria, fitting the PICO focused question”.

No restrictions concerning publication date nor status were applied in the eligibility criteria, as illustrated in Table 1. The small number of studies, totally five, considered in our overview is mainly ascribable to the inclusion criteria allowing for considering only systematic reviews exclusive comprising RCTs (and excluding observational studies), aiming to try to reach the higher level evidence. Secondarily, those systematic reviews comprising RCTs evaluating antibiotic administrations to control endodontic pain, as well as pre- and intra-operative medications administrations and endodontic treatments not completed, were also not taken into account in our overview, further reducing the number of the included studies.

Concern of the Referee: Why use antibiotics for pain control: "antibiotic treatment to control post-operative endodontic pain" ? 

Our response: We especially thank you for the consideration. Unfortunately, both in clinical practice and in scientific literature, antibiotic administration is often combined to NSAIDs and analgesic use, as pre-, intra- or postoperative medication, to control endodontic pain, against currently accepted standards for antibiotic stewardship and American Dental Association and American Association of Endodontists indications with regard to endodontic treatment. For such a reason, we included in the text the following period “the presented overview did not consider either pre-operatively administered oral medications or combined antibiotic prescriptions, in the attempt to highlight the shorter therapy duration, so excluding pre-medications, and to avoid unnecessary antibiotic treatment to control post-operative endodontic pain. Moreover, we aimed to identify most effective type and dosage of post-operatively administered medications, not only limiting at the most therapy duration and complying with accepted standards for antibiotic stewardship [48,69,70], but also taking into account polypharmacy, which is especially relevant for patients’ comorbidity [72,73] and specific conditions”. See also “Moreover, several reviews also evaluated antibiotics and pre-medications to control post-endodontic pain, which were both excluded from the present overview, because contrasting with antibiotic stewardship standards proposed both by the American Dental Association [63] and the American Association of Endodontists [64] and with the attempt of avoiding polypharmacy [64], respectively, and, should, therefore, be considered not clinically advisable”. Accordingly, given the same considerations, one article entitled “Effects of antibiotic administration on post-operative endodontic symptoms in patients with pulpal necrosis: A systematic review and meta-analysis” (doi:10.1111/joor.13057)has been excluded from the present overview, as illustrated in Figure 1.

  1. Concern of the Referee: This is a supposition: "potentially requiring additional operative sessions, that should be even more avoided in the ongoing context of the COVID-19" Covid 19 has nothing to do with pain evolution. 

Our response: We thank you for the suggestion. We have tried to re-phrase the period in order to make it more readable “In such a perspective, the correct pharmacological strategy should also reduce patients concern, on the one side, and optimize practitioners time and efforts, on the other side, thus decreasing endodontic complications and emergencies, which may potentially require additional operative sessions, that should be even more avoided in the ongoing context of the COVID-19”.

  1. Concern of the Referee: The practitioner intends to perform the root canal treatment correctly.... :"when correctly performed"  So, if pain emerges, it is not iatrogenically  intended.

Our response: We thank you for the suggestion. We have tried to re-phrase the period, in order to emphasize the potential iatrogenic damage and related pain, potentially due to incorrectly performed root canal treatment “Pain following endodontic treatment may be secondary to anatomical factors, such as apical foramen position, pulp tissue localization in areas where cannot be removed without proper care, and root canals that can be omitted without instrumentation, as well as to technical factors, affecting or occurring during the procedure, such as an inaccurate determination of the working length. The last could, in turn, lead to an excessive instrumentation, with extrusion of root canal debris beyond the apical foramen during instrumentation, an extension of the filling beyond the apex, and an incorrect use of irrigants, specifically sodium hypochlorite and hydrogen peroxide, potentially causing iatrogenic periapical discomfort [50,51]”.

Concern of the Referee: Re-consider "For effective endodontic pain management, the causative factors and treatment methods discussed above should be given due consideration to help the patient get rid of the unpleasant feeling of pain." "Presented recommendations" -  which? 

Our response: We thank you for the suggestion. We referred to the recommendations given in paragraph 4.1 - Oral medications post-operatively administered to control post-endodontic pain “Nonsteroidal anti-inflammatory drugs resulted the most common oral medicaments post-operatively administered to control pain, with ibuprofen being the most prescribed and investigated. Ibuprofen (600 mg) alone and ibuprofen (600 mg) combined with acetaminophen (1000 mg) was reported to be significantly more effective in post-endodontic pain control, when administered 6 hours after endodontic treatment, and may be, therefore, recommended as a first choice treatment in the first hours following non-surgical endodontic treatment. In addition, emerging data support the evidence that ketoprofen (50 mg) as well as naproxen (500 mg) might be even more effective than ibuprofen (600 mg) alone 6 hours post-operatively [47]. However, only low to moderate evidence support such a recommendation [48]. Moreover, multiple-dose regimens seemed to be more effective in pain control compared to single dose, although no definite protocols were proposed [45]”.

As suggested, in order to make the period more readable, we have modified it as follows “described clinical protocols, optimizing post-endodontic pain control, should be integrated with presented recommendations on post-operatively administered oral medicaments, proposing, the use of ibuprofen (600 mg) alone and combined with acetaminophen (1000 mg), as a first choice treatment, or ketoprofen (50 mg) as well as naproxen (500 mg), 6 hours post-operatively [47], also in the perspective of minimizing drug intake, as it may be especially relevant in fragile and elderly subjects”.

Concern of the Referee: Conclusions should be more precise. They are ambiguous. 

Our response: We especially thank you for the comment. Accordingly, we have tried to improve conclusion section, based on your suggestions, and we have highlighted changes in yellow.

Concern of the Referee: "fragile and elderly subjects" were not the subject of the research - so how can you conclude this? 

Our response: We thank you for the comment. Adulthood and old age were inclusion criteria, therefore, comorbidities and polypharmacy, relevant in the elderly and fragile or special needs patients, can not be ignored when prescribing additional medicaments, especially when systemically delivered. Accordingly, the aim of the present overview was “to  identify most effective type and dosage of post-operatively administered medications, not only limiting at the most therapy duration and complying with accepted standards for antibiotic stewardship [48,69,70], but also taking into account polypharmacy, which is especially relevant for patients’ comorbidity [72,73] and specific conditions”, as specified in the discussion section.

Reviewer 2 Report

This is a potentially interesting study concerning post-endodontic pain and its' management in terms of administered oral medications and integrated evidence-based clinical recommendations. As authors swiftly noticed, the problem is undervalued - yet management of pre- intra- and post-endodontic symptoms might lead to local and central sensitization. These processes may lead to increased risk of pain chronicization, if not maintained properly. Poorly managed pre- and post-operative pain is unfortunately very common yet preventable issue, so I read an entire manuscript with a great interest. Considering the multitude technical and pharmacological approaches proposed by authors to control post-endodontic pain, the study mainly focuses on summarizing findings about administered oral medications for post-endodontic pain control in its' narrative part. Authors also tried to evaluate the most effective type and pinpoint the most useful dosage of certain medicaments. Secondarily, in a part where narrative review is written - authors emphasized current evidence on technical solutions intertwined with endo procedures, to increase pain control and improve post-endodontic pain management.

Study itself in form as it is written and conducted lacks systematic-review and meta-analysis of RCTs - which is a major flaw of this paper, and renders the article unpublishable at this point. Certain tools used (PRISMA, PICO etc) are valid mostly for SR's so it is quite surprising, that authors did not manage it into SR with MA. Minor remarks below:

As authors state in the title study is focused on 'post-endodontic pain', not 'post-operative pain' or 'post-operative endodontic pain either' - there is no such thing like latter in Orofacial Pain classification. Post-operative pain is also diffferent by definition. Please change this accordingly in the abstract section and throughout entire manuscript for clarity.  

Introduction

This section is full of unrelated and barely-related self-citations.

L46-53 - there is no mention about chronic pain and sensitization possibility in this section, please add a few words about it, with relevant ciation(s) - I suggest 

  1. Al-Sabbagh M, Okeson JP, Bertoli E, Medynski DC, Khalaf MW. Persistent pain and neurosensory disturbance after dental implant surgery: prevention and treatment. Dent Clin North Am. 2015 Jan;59(1):143-56. doi: 10.1016/j.cden.2014.08.005. Epub 2014 Sep 22.
  2. Ehsani M, Moghadamnia AA, Zahedpasha S, Maliji G, Haghanifar S, Mir SM, Kani NM. The role of prophylactic ibuprofen and N-acetylcysteine on the level of cytokines in periapical exudates and the post-treatment pain. Daru. 2012 Sep 10;20(1):30. doi: 10.1186/2008-2231-20-30.
  3. Dalewski B, Kamińska A, Szydłowski M, Kozak M, Sobolewska E. Comparison of Early Effectiveness of Three Different Intervention Methods in Patients with Chronic Orofacial Pain: A Randomized, Controlled Clinical Trial. Pain Res Manag. 2019 Mar 11;2019:7954291. doi: 10.1155/2019/7954291.

Materials and methods

L73 -  'statement42' ?

L77 - 'model43' ?

L82-88 and L139-145 PICO statements are basically the same - please either unify or remove the one redundant

Results

This section is very messy, tables should be re-written for clarity

Discussion

L287-288 - 'and to avoid unnecessary antibiotic treatment to control post-operative endodontic pain' - antibiotics are not analgesics - either specify with relevant citation(s) or remove if redundant

To sum up, the narrative  part of the article is quite good and informative, hower lacks scientific value as it counts as an opinion. And as such after some extension and editing might be more than suitable onto some non-peer reviewed  journal.

Author Response

On behalf of the Authors, I would first like to express our appreciation for the time and efforts spent reviewing our manuscript and for the comments and suggestions offered by the Referee and apologize for the difficulties encountered.

We welcome the suggestions for improving manuscript readability and we have tried to address the concerns expressed.

The authors have made changes to the manuscript according to the valuable comments and suggestions and to assist you in your work, we have highlighted all changes to the manuscript in blue and we have reported the point-to-point replies to questions, concerns and suggestions below.

Table 1 and Table 2 have been edited and correctly located in the text.

We sincerely hope that the man script is now suitable for publication.

Kind regards

Dr. Federica Di Spirito, DDS, PhD

Concern of the Referee: Study itself in form as it is written and conducted lacks systematic-review and meta-analysis of RCTs - which is a major flaw of this paper, and renders the article unpublishable at this point. Certain tools used (PRISMA, PICO etc) are valid mostly for SR's so it is quite surprising, that authors did not manage it into SR with MA.

Our response: We thank you for the comments. The present study comprises both an overview of systematic reviews on post-operatively administered oral medications, exclusively including RCTs (allowing us to apply PRISMA, PICO and AMSTAR2 tools), and a narrative review on proposed technical solutions, to optimally control post-operative endodontic pain.

Concerning the overview of systematic review, it has been renamed throughout the manuscript to emphasize the fundamental process of synthesising evidence which is derived, often exclusively, from systematic reviews. The systematic review forms the primary ‘unit of analysis’ and is the basis upon which an overview is built (Hunt, H., Pollock, A., Campbell, P., Estcourt, L., & Brunton, G. (2018). An introduction to overviews of reviews: planning a relevant research question and objective for an overview. Systematic reviews, 7(1), 39. https://doi.org/10.1186/s13643-018-0695-8).

We also welcomed an indirect suggestion and, regarding the overview of systematic reviews, a meta-analysis could not be, unfortunately, performed, as stated in the newly added period in Result section “No meta-analysis could be performed due to the high heterogeneity and incompleteness of retrieved data”.

Concern of the Referee: As authors state in the title study is focused on 'post-endodontic pain', not 'post-operative pain' or 'post-operative endodontic pain either' - there is no such thing like latter in Orofacial Pain classification. Post-operative pain is also diffferent by definition. Please change this accordingly in the abstract section and throughout entire manuscript for clarity.  

Our response: We especially thank the Referee for such a consideration and suggestion and we have modified the expression in “post-operative endodontic pain” throughout the manuscript.  

Concern of the Referee: there is no mention about chronic pain and sensitization possibility in this section, please add a few words about it, with relevant citation(s).

Our response: Once again, we especially thank the Reviewer for such a comment, enhancing study background. We have briefly discussed to topic “Post-operative endodontic pain remains still often undervalued, although it may lead, if not correctly managed, to both local and central sensitization, with pain chronicization” and added suggested references, along with Pigg, M., Nixdorf, D. R., Law, A. S., Renton, T., Sharav, Y., Baad-Hansen, L., & List, T. (2021). New International Classification of Orofacial Pain: What Is in It For Endodontists?. Journal of endodontics47(3), 345–357. https://doi.org/10.1016/j.joen.2020.12.002.

Concern of the Referee: L73 -  'statement42' ? L77 - 'model43' ?

Our response: We apologize for the mistake, it was a typo error (square bracket missing).

Concern of the Referee: This section is very messy, tables should be re-written for clarity

Our response: We apologize for the inconvenient maybe due to Journal editing; we have correctly located Table 1 after the correct paragraph and re-inserted Table 2.

Concern of the Referee: 'and to avoid unnecessary antibiotic treatment to control post-operative endodontic pain' - antibiotics are not analgesics - either specify with relevant citation(s) or remove if redundant

Our response: We especially thank you for the consideration. Unfortunately, both in clinical practice and in scientific literature, antibiotic administration is often combined to NSAIDs and analgesic use, as pre-, intra- or postoperative medication, to control endodontic pain, against currently accepted standards for antibiotic stewardship and American Dental Association and American Association of Endodontists indications with regard to endodontic treatment. For such a reason, we included in the text the following period highlighted in yellow “the presented overview did not consider either pre-operatively administered oral medications or combined antibiotic prescriptions, in the attempt to highlight the shorter therapy duration, so excluding pre-medications, and to avoid unnecessary antibiotic treatment to control post-operative endodontic pain. Moreover, we aimed to identify most effective type and dosage of post-operatively administered medications, not only limiting at the most therapy duration and complying with accepted standards for antibiotic stewardship [48,69,70], but also taking into account polypharmacy, which is especially relevant for patients’ comorbidity [72,73] and specific conditions”. See also “Moreover, several reviews also evaluated antibiotics and pre-medications to control post-endodontic pain, which were both excluded from the present overview, because contrasting with antibiotic stewardship standards proposed both by the American Dental Association [63] and the American Association of Endodontists [64] and with the attempt of avoiding polypharmacy [64], respectively, and, should, therefore, be considered not clinically advisable”. Accordingly, given the same considerations, one article entitled “Effects of antibiotic administration on post-operative endodontic symptoms in patients with pulpal necrosis: A systematic review and meta-analysis” (doi:10.1111/joor.13057)has been excluded from the present overview, as illustrated in Figure 1.

Reviewer 3 Report

the very precise inclusion criteria led to the examination of only 5 articles but despite the limited number the results and conclusions are interesting. I agree with the exclusion of those articles in which pre-intervention therapy and post-intervention antibiotics were administrated . the  article was overall well written and precise, perhaps repetitive at times.

Author Response

On behalf of the Authors, I would first like to express our appreciation for the time and efforts spent reviewing our manuscript and for the comments and suggestions offered by the Referee and apologize for the difficulties encountered.

We welcomed the suggestions for improving manuscript readability and we have tried to address the concerns expressed.

We welcome the suggestions for improving manuscript.

Table 1 and Table 2 have been edited and correctly located in the text.

We sincerely hope that the man script is now suitable for publication.

Comment of the Referee: the very precise inclusion criteria led to the examination of only 5 articles but despite the limited number the results and conclusions are interesting. I agree with the exclusion of those articles in which pre-intervention therapy and post-intervention antibiotics were administrated . the  article was overall well written and precise, perhaps repetitive at times.

Our response: We especially thank you for the consideration and, in accord with your observation regarding the small number of included articles, we have stated in the manuscript, along with other study limitations, “The main limitation of this overview is the small number of included studies, ascribable to the restricted eligibility criteria, fitting the PICO focused question”.

No restrictions concerning publication date nor status were applied in the eligibility criteria, as illustrated in Table 1. The small number of studies, totally five, considered in our overview is mainly ascribable to the inclusion criteria allowing for considering only systematic reviews exclusive comprising RCTs (and excluding observational studies), aiming to try to reach the higher level evidence. Secondarily, those systematic reviews comprising RCTs evaluating antibiotic administrations to control endodontic pain, as well as pre- and intra-operative medications administrations and endodontic treatments not completed, were also not taken into account in our overview, further reducing the number of the included studies.

We have tried to address the concern expressed, editing the manuscript.

Round 2

Reviewer 1 Report

None

Reviewer 2 Report

An entire manuscript has greatly improved, I feel it is publication ready now.